# Dental Pulp Response to Different Types of Calcium-Based Materials Applied in Deep Carious Lesion Treatment—A Clinical Study

**DOI:** 10.3390/jfb13020051

**Published:** 2022-05-02

**Authors:** Antoanela Covaci, Lucian Toma Ciocan, Bogdan Gălbinașu, Mirela Veronica Bucur, Mădălina Matei, Andreea Cristiana Didilescu

**Affiliations:** 1Department of Embryology, Faculty of Dental Medicine, “Carol Davila” University of Medicine and Pharmacy, 050474 Bucharest, Romania; antoanela.covaci@ugal.ro (A.C.); Andreea.Didilescu@umfcd.ro (A.C.D.); 2Department of Dental Medicine, Faculty of Medicine and Pharmacy, Dunarea de Jos University of Galati, 800010 Galati, Romania; Madalina.Matei@ugal.ro; 3Department of Prosthetics Technology and Dental Materials, Faculty of Dental Medicine, “Carol Davila” University of Medicine and Pharmacy, 010221 Bucharest, Romania; mirela.bucur@umfcd.ro

**Keywords:** pulp capping, dental, calcium hydroxide, silicates

## Abstract

Dental pulp vitality preservation in dental caries treatment is a major goal in odontotherapy. The main objective of this study was to compare dental pulp tissue responses to vital therapies in deep carious lesions, using different calcium-based materials. An ambispective study was conducted on 47 patients. Ninety-five teeth with deep carious lesions were treated. Among them, 25 (26.32%) were diagnosed with pulpal exposures and treated by direct pulp capping. Indirect pulp capping was applied when pulp exposure was absent (n = 70; 73.68%). Fifty teeth (52.63%) were treated with TheraCal LC (prospective study), 31 teeth (32.63%) with Calcimol LC, and 14 teeth (14.74%) with Life Kerr AC (retrospective study). The results show that the survival rate for dental pulp was 100% for Life Kerr AC, 92% for TheraCal LC, and 83.87% for Calcimol LC, without significant differences. Apparently, self-setting calcium hydroxide material provided better dental pulp response than the two light-cured materials, regardless of their composition, that is, either calcium -hydroxide or calcium silicate-based. We will need a significant number of long-term clinical studies with the highest levels of evidence to determine the most adequate biomaterials for vital pulp therapies.

## 1. Introduction

Dental pulp vitality preservation in dental caries treatment is a major goal in odontotherapy. Both modern dentistry and extended histological research conclude and accept that nothing can replace the dental pulp, generating the same benefits for the tooth and the periodontium [1].

Dental caries is a disease caused by multiple factors and involves interactions of three factors: the body of the host, represented primarily by the teeth and the saliva; the diet, determined by the availability of fermentable carbohydrates; and the microbiota, which are acid-producing bacteria [2].

Depending on their position, depth, or extension, caries generates major difficulties in achieving the correct technique of classic treatment. Sometimes, the occult evolution (by the interproximal position) in the absence of a careful clinical/radiological exam, makes pathology detection impossible before the irreversible pulp disease. Rapidly progressive or slowly progressive lesions inevitably converge toward pulp damage, so, in both situations, the treatment must be rapid, complex, and rigorous [3].

Ever since the mid-1970s, studies have indicated that the pulp tissue can tolerate different dental restorative materials as long as bacteria and their toxins can be excluded from the pulp tissue [4]. This is the goal of direct/indirect capping.

Calcium hydroxide was introduced in dentistry in 1921 and has been considered the “gold standard” of direct pulp-capping materials for many years. [5] This material is considered to have excellent antibacterial properties [6], and one of the studies found a complete reduction of the micro-organisms that are frequently associated with pulp infections after only one hour of contact with calcium hydroxide [7]. More than that, calcium hydroxide has one of the best clinical success rates and long-term follow-up rates as a pulp-capping agent after different periods, even after 10 years [8].

But calcium hydroxide is not infallible. The self-cure formulations are highly soluble and can dissolve in time [9], but it has been noticed that by the time the calcium hydroxide disappears because it dissolves, new bridges of detin are formed [8,10]. It provides a poor seal [11] and has no inherent adhesive qualities. Another concern about this material would be the appearance of “tunnel defects” in reparative dentin formed underneath calcium hydroxide pulp caps [12].

MTA is primarily composed of calcium oxide in dicalcium silicate, tricalcium silicate, and tricalcium aluminate form. Bismuth oxide can be added for its radiopacity effect [13]. MTA is considered a silicate cement rather than an oxide mixture, and so its biocompatibility is based on its reaction products [14]. It is important to note that the primary reaction product of MTA with water is calcium hydroxide [15], and so calcium hydroxide’s formation is actually the one that provides MTA’s biocompatibility, so they are rather similar. However, a significant difference would be the fact that MTA provides some seal to tooth structure [16].

An important downside to MTA manipulation and clinical use is the prolonged setting time; some products need more than 2 h [17]. This implies that pulp capping with MTA is clinically possible either using a quick-setting liner to protect the MTA during permanent restoration placement or performing a two-step procedure.

The main objective of the present study was to compare the clinical and biological effects of different calcium-based pulp-capping materials on dental pulp responses to vital therapies in deep carious lesions. The specific aim was to clinically assess and compare the pulp vitality, following the above-mentioned therapies.

## 2. Materials and Methods

This ambispective clinical study was conducted in compliance with the research ethics legislation currently in place in Romania. Informed consent was obtained from all subjects involved in the study. The study was conducted in accordance with the Declaration of Helsinki, and the protocol was approved by the Ethics Committee of Dunarea de Jos University of Galati (no. 4842/20/02/2020). The study was conducted by the same investigator (Antoanela Covaci).

### 2.1. Sample Selection

The medical files of patients who underwent direct or indirect pulp-capping therapies in Dr. Antoanela Covaci’s private practice from February 2017 to May 2019 consisted of the retrospective, control sample. Patients attending the private practice from 1 July 2020 to 20 December 2021 were included in the prospective study. Patients with direct/indirect pulp cappings with three dentinogenesis-inducing materials, Theracal LC (Bisco Inc., Schaumburg, IL, USA), Calcimol LC (Voco GmbH, Cuxhaven, Germany), and Life Kerr AC (Kerr, Orange, CA, USA), were selected to be included into the study.

#### Inclusion Criteria

The included patients met certain criteria, namely, thermal sensitivity response compatible with a diagnosis of tooth vitality and association of radiographic examination, a reasonable state of health and oral hygiene, and no associated periodontal pathology. The included treatments had the same protocol: isolation of the operative field with cotton rolls type 2 and cavity cleaned with neophaline. The hydrogen peroxide on sterilized cotton pellets was used for hemostasis in case of accidental pulp exposure. Teeth in which restorations were performed using self-etching adhesive systems such as Filltec, GC, Voco, Beautifil, and Flow were included. Light-cured composite resins were used as restorative materials in all the therapies.

### 2.2. Evaluation of the Clinical Procedures of Direct/Indirect Pulp Capping

Ninety-five direct/indirect pulp-capping therapies performed on 47 patients were included and divided into three groups: the first group with 50 teeth was treated with TheraCal LC (13 anterior teeth, 16 premolars, and 21 molars); the second group with 31 teeth was treated with Calcimol LC (12 anterior teeth, 9 premolars, and 10 molars), and the third group with 14 teeth was treated with Life Kerr AC (1 anterior tooth and 13 molars). The etiologies of the pulp exposures were different because of extensive carious lesions and accidentally, at the moment of removal, of soft, infiltrated dentin; therefore, different dimensions of the exposure sites were reported. In all clinical cases, the therapies were performed with one of the above-mentioned biomaterials, followed by the placement of direct resin or glass-ionomer restorations of the crowns. The permanent restorations were performed at the appointment in which the direct/indirect pulp capping was performed.

### 2.3. Evaluation of the Teeth after Direct/Indirect Capping Therapy

One month, 3 months, and 6 months after treatment, clinical controls were performed. During these periodical controls, qualitative tests were performed such as the pulp vitality test, which included the vertical and horizontal percussion tests (see Figure 1).

Therapies on teeth that remained asymptomatic, with normal sensitivity tests and no other radiographic signs, such as periapical pathology, were considered as clinical successes.

### 2.4. Statistical Analysis

Data distributions were expressed as means, standard deviations (SD) and percentages. Pearson’s chi-squared tests were used for categorical measures. When the expected frequency of any cell in the table was <5, Fisher’s exact test was used

Statistical analyses were performed using Stata/IC 16 (StataCorp. 2019. Stata Statistical Software: Release 16. StataCorp LLC.: College Station, TX, USA), and *p*-values < 0.05 were considered statistically significant.

## 3. Results

The study included 47 urban patients (70.21% females; mean age 34.66 ± 11.15 years). Ninety-five teeth, presenting deep carious lesions, were treated. Among them, 25 (26.32%) were diagnosed with pulpal exposures and treated by direct pulp capping. Indirect pulp capping was performed when pulp exposure was absent. Fifty teeth (52.63%) were treated with TheraCal LC (prospective study); 31 teeth (32.63%) were treated with Calcimol LC, and 14 teeth (14.74%) were treated with Life Kerr AC (retrospective study). Regarding direct pulp capping, 4 teeth were treated with Life Kerr AC, and 21 teeth were treated with Theracal LC. Indirect pulp capping was applied as follows: 10 teeth with Life Kerr AC, 31 teeth with Calcimol, and 29 teeth with TheraCal LC.

Features of the 95 carious lesions treated and analyzed are presented in Table 1. The prospective cohort comprised significantly more pulp exposures than the retrospective cohort (Table 1).

The results of tooth vitality preservation 6 months after treatment for the materials taken into the study are presented in Table 2.

## 4. Discussion

Taking into account the analysis of the results registered in the present study, we can draw multiple conclusions.

Regarding the specific aim of the study, the rate of preservation of pulp vitality was different for the three pulp-capping materials although the difference was not statistically significant (see Table 2). Among the only 14 teeth (15% of 95 teeth analyzed) treated with Life Kerr AC, none of them lost vitality after 6 months. For the other two materials, the vitality of the treated teeth was lost in 8% of the cases treated with TheraCal LC and in 16.13% of the cases treated with Calcimol LC.

The above-mentioned results were not related to the position of the affected tooth (anterior or posterior), and they seem not even related to the initial carious activity of the treated lesion. All materials for pulp capping taken into the study (TheraCal LC, Calcimol LC, and Life Kerr AC) are indicated to be used for pulp capping, although Calcimol LC is recommended only for indirect pulp-capping treatments.

At the time of cavity preparation, by getting close to pulp tissue or, even worse, exposing it using rotary instruments, the pulp tissues can become inflamed or necrotized. In this case, the clinician often makes the decision to perform an endodontic treatment. For this reason, the materials indicated for pulp capping should act as a barrier and protect the vitality of the entire pulp by covering the minimally exposed tissue and preventing the need for further endodontic treatments. At the same time, the capping material used should prompt a regenerative response from the host side [18].

TheraCal LC (Bisco Inc., Schaumburg, IL, USA) is a light-cured, resin–calcium silicate matrix. This liner material is recommended for direct and indirect pulp capping. It contains monomers of polymerizable methacrylate, Portland cement type III, polyethylene glycol dimethacrylate, and barium zirconate [19].

Calcimol LC (Voco GmbH, Cuxhaven, Germany) is a light-cured, pulp-capping material. This material represents a resin-modified calcium ion-releasing liner. In this material, the calcium dihydroxide is embedded in a resin–polymethacrylate matrix: urethane dimethacrylate, dimethylaminoethyl-methacrylate, and triethyleneglycol dimethacrylate (TEGDMA) [20].

Life Kerr AC (Kerr GmbH, Karlsruhe, Germany) is a self-setting, calcium ion-releasing liner and pulp-capping material. It contains calcium dihydroxide, N-ethyl-o(or p)-toluenesulphonamide, zinc and calcium oxide, methyl salicylate, and 2,2-dimethylpropane-1,3-diol [21].

Light-curable, resin-modified calcium hydroxide materials, such as two of the materials taken into the study, are TheraCal LC (Bisco) and Calcimol LC (Voco), largely used for direct pulp capping. Compared to the conventional two-paste calcium hydroxide systems, the third material taken into the study, Life Kerr (Kerr), the resin-modified versions have several advantages, including ease of handling, light polymerization, and superior physical properties. They are also minimally affected by phosphoric acid and have low water solubility, which means that they do not dissolve in time.

Despite the multiple advantages of the resin-modified calcium hydroxide liners, due to the light-activated polymerization, we can say that there is a significant risk of free residual monomers left at the pulp-capping site. It is known that unpolymerized monomers are toxic to pulp cells [22]. For example, Calcimol LC was reported to present higher cytotoxicity to MDPC-23 cells than another resin-free calcium hydroxide paste. However, the composite resin is considered to present mild to no toxic effects to the odontoblast-like MDPC-23 cells if it is polymerized [23]. Another study shows that resin-modified calcium hydroxide is not more cytotoxic than the control calcium hydroxide. When light resin-modified pulp-capping materials with light-activated polymerization are sufficiently cured, with a longer curing time, the cytotoxicity effect of the resin disappears. Meanwhile, in the time of polymerization, OH− is released and can cause some cytotoxicity [24]. We noticed that a mean rate of 70% conversion of the polymerization in the case of dimethacrylate monomers does not mean the presence of 30% unreacted free monomers. It means that only 30% of the methacrylate groups remains still active for polymerization, but among those, most of them are already inside the polymer matrix. Overall, only a small percentage (9% of monomers) can be considered free. (i.e., both methacrylate groups in one monomer are not cured), and most of these free monomers are located inside the polymer matrix (cannot be released) [25]. There are some studies [26,27] that report that cytotoxicity was not observed in the MG63 cells treated with TheraCal LC, and after 5 days, the cells are organized as a confluent monolayer as demonstrated by fluorescence microscopy observations. Theracal LC shows biocompatibility on MG63 cells allowing physiological cell growth and differentiation. Chemical and physical properties and Theracal LC biocompatibility observed in in vitro studies still consider this cement as an efficient pulp-capping material for the vital pulp therapy [27].

From a clinical point of view, there is a difference between the two resin-modified pulp-capping materials taken into the study. Calcimol LC is easier to handle and more tooth-colored compared to the opaque white of TheraCal LC. In addition, taking into consideration the manufacturer’s indications, Calcimol LC can be used with or without dental adhesives [28,29]. Materials with new compositions are evaluated comprehensively before their clinical application. There are recent studies that examined whether the lower calcium ion-releasing ability, together with the cytotoxicity because of unpolymerized resin monomers of resin-modified calcium ion-releasing liners, has an influence on its biological and clinical performance [30]. The pH values evaluated in another study [31] were slightly alkaline for TheraCal LC and Calcimol LC, compared to the control group. The lower pH values registered were because the ions were dispersed through the different dentine thicknesses that remained. This discovery contrasts with other studies [32,33]. Approaching a physiological pH within 60 days during this study may provide a positive environment for pulpal cell viability and metabolic movement with the reparative dentine development. TheraCal LC has been demonstrated to discharge higher Ca2+ ions right after application and to make a natural pH, very close to the physiological one in the first two months. Further clinical tests are required to measure the release of different biologically active ions from TheraCal LC, which can contribute to the clinical success of these materials in vital pulp therapies [31,32,34].

Another randomized systematic review study, analyzing long-term clinical and radiographic evaluation of the effectiveness of direct pulp capping materials, showed that multiple variables must be taken into consideration for an accurate interpretation of pulp-capping material’s efficiency [35]. The aim of this review was to assess the effectiveness of 12 different direct pulp capping materials for dental pulp exposures. Long-term clinical and paraclinical (X-rays) evaluations of the efficiency of different direct pulp-capping materials used on teeth with pulp exposure were included. After a risk of bias assessment and data acquisition and interpretation from 496 identified articles, only 15 met the eligibility criteria. From all the studies that were included in those articles, a total of 1322 teeth were treated with 12 different types of direct pulp-capping materials. However, the results were based on the present studies, which were all judged to have a high risk of misinterpretation. In this evaluation, many materials were studied, and some of them seemed to perform better than calcium hydroxide materials, as for example Life Kerr (Kerr). However, unlike calcium hydroxide, all the other materials were supported by only a small number of studies. Therefore, more long-term clinical and radiographic studies with lower a risk of bias are needed [35].

Although calcium hydroxide (as Life Kerr) has long been considered the gold standard for direct pulp capping, it has some disadvantages: The high pH can irritate the dental pulp and can cause the inflammation or the necrosis of the exposed pulp surface. The newly formatted dentine can have tunnel-shaped defects, and the dissolution in time may lead to failure of the long-term seal. All these disadvantages are probably responsible for the wide differences in success rates, ranging from 52% to 100% [35,36,37].

The differences in the protocols of isolation and antiseptisation are an important factor that must be taken into consideration for the accurate interpretation of different studies. In another study with similar conditions of isolation and antiseptisation, a study in which 69 teeth were treated with calcium hydroxide, 57 received indirect pulp capping, with 53 (93.0%) showing a successful outcome and 4 (7.0%) an unsuccessful outcome [38]. This result can explain the efficiency of 100% of calcium hydroxide that was analyzed in our study and contributed to the smaller number of teeth (only 14) involved.

The limits of the study include the design and relatively small sample size. Because of the restrictions imposed by COVID-19, patient recruitment availability was seriously affected, and we could not provide similar sample sizes for the materials that were investigated. It is obvious that the retrospective component of the study relied mostly on the accurate recordkeeping. However, considering that the same investigator was involved in all treatments, we consider that the results of the study were less biased.

More than 20 types of biomaterials are effective in direct pulp capping. Until now, an ideal pulpotomy material has not been established [39,40,41]. A greater number of long-term clinical studies with highest levels of evidence (randomized control tests) are required to determine the best composition biomaterial for direct or indirect pulp capping.

## 5. Conclusions

Within the limits of the study, our results suggest a better preservation of dental pulp vitality in the case of self-setting calcium hydroxide Life Kerr AC (Kerr) as compared to resin-modified calcium-releasing TheraCal LC (Bisco) and Calcimol LC (Voco). Among the two light-activated materials, TheraCal LC (Bisco) and Calcimol LC (Voco), TheraCal LC seems to have a better potential in keeping pulp vitality, probably due to a better local pH maintenance.

A greater number of long-term clinical studies with the highest levels of evidence are needed to determine the most adequate biomaterials for vital pulp therapies. Moreover, extensive research has to be carried out to improve dental materials in order to maintain the dental pulp potential and facilitate its regeneration in the case of cavities aggression.

## Figures and Tables

**Figure 1 jfb-13-00051-f001:**
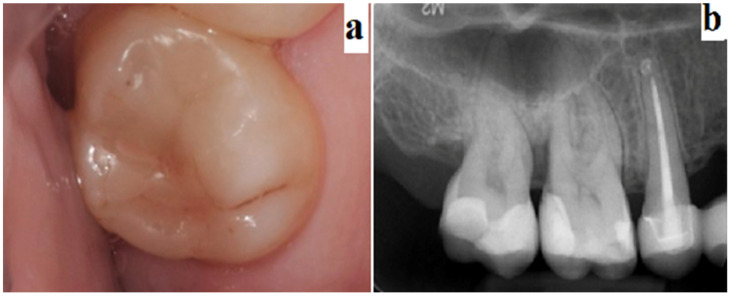
Example of evaluation 6 months post-op: Clinical evaluation—marginal infiltration (**a**); periapical X-ray evaluation (**b**).

**Table 1 jfb-13-00051-t001:** Characteristics of the carious lesions treated.

	Retrospective Cohort (n = 45)		Prospective Cohort (n = 50)		*p*
	n	%	n	%	
**Pulpal exposure**					
Yes	4	8.89	21	42	<0.001
No	41	91.11	29	58	
**Capping material**					
TheraCal LC	0	0	50	100	<0.001
Calcimol LC	31	68.89	0	0	
Life Kerr AC	14	31.11	0	0	
**Affected teeth**					
Anterior	13	28.89	13	26	0.753
Posterior	32	71.11	37	74	
**Lesion activity**					
Active	4	8.89	0	0	0.572
Arrested	41	91.11	50	100	

**Table 2 jfb-13-00051-t002:** Outcome assessment.

Capping Material	Vitality Preservation		*p*
	No	Yes	
TheraCal LC	4 (8%)	46 (92%)	0.236
Calcimol LC	5 (16.13%)	26 (83.87%)	
Life Kerr AC	0	14 (100%)	

## Data Availability

The data presented in this study are available on request from the corresponding author.

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
