# Peer review of "Dental Pulp Response to Different Types of Calcium-Based Materials Applied in Deep Carious Lesion Treatment—A Clinical Study"

_jfb, 2022, doi:10.3390/jfb13020051_

Round 1

Reviewer 1 Report

In this study, the authors described that the main objective of the present study was to compare the effects on dental pulp response to vital therapies in deep carious lesions, using different calcium-based materials. An ambispective study was conducted on 47 patients. Ninety-five teeth, presenting deep carious lesions, were treated. Among them, 25 (26.32%) were diagnosed with pulpal exposures and treated by direct pulp capping. Indirect pulp capping was performed when pulp exposure was absent. Fifty teeth (52.63%) were treated with TheraCal LC (prospective study), 31 teeth (32.63%) with Calcimol LC and 14 teeth (14.74%) with Life Kerr AC, respectively (retrospective study). The results showed that the survival rates for dental pulp were 100% for Life Kerr AC, 92% for TheraCal LC, and 83.87% for Calcimol LC, without statistically significant differences. Self-setting calcium hydroxide material seamed to provide better dental pulp response than the two light-cured materials, irrespective of their composition, that is, either calcium-hydroxide or calcium-silicate-based. The sample size of  materials compared were different.  As the authors also mensioned that greater number of long-term clinical studies with highest levels of evidence are needed to determine the most adequate biomaterials for therapies. In general, this study provided some information and could be better if the sample size of different materials  were similar for comparison. 

Reviewer 2 Report

Dear Authors,

I read your manuscript, the topic is interesting, but some issues need clarification.

Best regards
